# Upregulated Hexokinase-2 in Airway Epithelium Regulates Apoptosis and Drives Inflammation in Asthma via Peptidylprolyl Isomerase F

**DOI:** 10.3390/cells14131004

**Published:** 2025-07-01

**Authors:** Zhen Tian, Hongyan Zheng, Yan Fan, Boyu Li, Zhenli Huang, Meijia Wang, Jixian Zhang, Jianping Zhao, Shanshan Wang, Jungang Xie

**Affiliations:** 1Department of Respiratory and Critical Care Medicine, National Clinical Research Center of Respiratory Disease, Key Laboratory of Pulmonary Diseases of Health Ministry, Tongji Hospital, Tongji Medical College, Huazhong University of Science and Technology, Wuhan 430030, China; t1074904364@gmail.com (Z.T.); zhenghy814@163.com (H.Z.); fanyan9612@163.com (Y.F.); lby199804@163.com (B.L.); zlhuang163@163.com (Z.H.); meijia.wang@hotmail.com (M.W.); zhaojp88@126.com (J.Z.); 2Department of Respiratory and Critical Care Medicine, Hubei Province Integrated Traditional Chinese and Western Medicine Hospital, Wuhan 430015, China; jxzhang1607@163.com

**Keywords:** asthma, hexokinase 2, glycolysis, airway inflammation, apoptosis

## Abstract

Hexokinase catalyzes the first rate-limiting step glycolysis. However, the roles of hexokinase 2 (HK2) in asthma remain incompletely understood. This study aimed to investigate metabolic alterations in asthma, focusing on the expression, function and regulation of HK2. In this study, non-targeted metabolomics analysis of 20 asthma patients and 15 healthy controls identified metabolic alterations in asthma, particularly in the glycolytic pathways. Consistently, HK2 expression was elevated in both asthma individuals and mice with allergic airway inflammation. Airway epithelium–specific HK2 knockdown and pharmacological inhibition with 2-deoxy-D-glucose (2-DG) significantly attenuated airway inflammation and hyperresponsiveness in mice induced by ovalbumin/ lipopolysaccharide. Mechanistic analyses demonstrated that HK2 regulates epithelial apoptosis and inflammation via interaction with peptidylprolyl isomerase F (PPIF), independent of voltage-dependent anion channel 1 (VDAC1). Asthma is associated with metabolic reprogramming, characterized by alterations in lipid and glucose metabolism. These findings establish HK2 plays a crucial role in asthma pathogenesis by promoting airway epithelial apoptosis and inflammation in asthma, suggesting its potential as a therapeutic target.

## 1. Introduction

Asthma is a chronic respiratory disorder marked by persistent inflammatory changes, airway hyperresponsiveness (AHR), and airway structural alterations [1,2,3]. Despite substantial efforts to improve living conditions and socioeconomic factors for asthma, its prevalence continues to rise [4,5,6]. Metabolic reprogramming has emerged as a critical contributor to asthma pathogenesis by regulating immune cell activation, differentiation, and effector functions. These metabolic alterations sustain chronic inflammation, promote airway remodeling, and contribute to epithelial dysfunction, thereby exacerbating disease severity and persistence. Individuals with asthma are more susceptible to obesity, which is associated with disruptions in glucose and lipid metabolism [7,8]. Several mechanisms linking metabolic disorders to asthma have been identified [9,10], Nevertheless, the complex interplay between asthma and metabolic dysregulation remains incompletely understood and warrants further investigation. Given the emerging interest in targeting metabolic pathways as therapeutic strategies in inflammatory diseases, elucidating the metabolic regulation in airway epithelium may provide novel insights into asthma pathogenesis and potential intervention approaches.

Airway epithelial dysfunction, triggered by allergens, pollutants, and viral or bacterial insults, is a key contributor to asthma pathogenesis by driving inflammation, chemotaxis, remodeling, and antimicrobial defense. These processes, in turn, further impair epithelial barrier function [11,12,13]. Importantly, airway epithelial cells are actively involved in orchestrating immune responses across different asthma endotypes, including both type 2 (T2-high) and non-type 2 (T2-low) forms, which reflect underlying heterogeneity in disease mechanisms [14]. Abnormalities in cellular metabolism, such as mitochondrial dysfunction, altered glucose and lipid metabolism, and oxidative stress, have been implicated in exacerbated airway inflammation and remodeling in asthma [15,16,17,18,19]. While glycolytic reprogramming is known to regulate immune cell activation and differentiation [15], its role in structural cells, such as the airway epithelium, remains largely unexplored.

Hexokinases catalyze the first rate-limiting step in glucose metabolism, by transforming glucose into glucose-6-phosphate [20,21]. Among the five hexokinase isoforms, hexokinase 2 (HK2) is the predominant form due to its high glucose affinity and dual functional catalytic domains [20,22]. Elevated HK2 has been implicated in various cancers, including lung, breast, colorectal cancers, and glioblastoma [22,23], as well as inflammatory diseases, such as colitis [24] and rheumatoid arthritis [25,26,27]. In activated T cells, while HK2 is upregulated, its deficiency does not impair the development of airway inflammation [28,29]. The role of HK2 in asthma is worth exploring.

In this study, we investigated metabolic reprogramming in asthma, with a specific focus on glycolysis. Given the observed elevation of HK2 expression in both asthma patients and murine models, we developed a mouse model with airway epithelium-specific HK2 deficiency to explore its role in asthma pathogenesis. Additionally, the HK2 inhibitor 2-DG was utilized to further validate the effects of HK2 in asthma.

## 2. Materials and Methods

### 2.1. Human Subjects

A total of 24 patients with asthma and 20 healthy subjects (HCs) were recruited in our hospital. The clinical data of enrolled participants is provided in Appendix A. There were no significant differences in sex distribution, age, or body mass index between asthma patients and healthy controls. Compared to healthy controls, asthma patients demonstrated reduced lung function. In addition, serum total immunoglobulin E levels and blood eosinophil counts were significantly elevated in the asthma group. Blood samples were obtained and processed as previously described by Wang et al. [30]. In brief, peripheral blood mononuclear cells (PBMC) and plasma were isolated from blood specimens. For plasma analysis, 300 μL of methanol, which contained 2-Chloro-L-phenylalanine (internal standard; 5 μg/mL), was added to 200 μL of each plasma sample. Following that, the mixture was centrifuged for 10 min at 4 °C at 13,000 rpm, and 200 μL of the supernatants were transferred to sampler vials for in-house quality control. An in-house quality control (QC) was prepared by mixing an equal volume of each sample. LC-MS analysis was conducted using the Agilent 1290 Infinity II UHPLC system coupled to Accurate-Mass Q-TOF/MS and Agilent 6545 UHD (Agilent Technologies, Santa Clara, CA, USA).

Endobronchial biopsies were obtained from lung nodules patients without asthma undergoing lobectomy or segmentectomy (controls; *n* = 8) and from asthma patients (*n* = 7) while undergoing bronchial bronchoscopy. The sample size was determined based on the number of eligible patients accessible during the study period. Given the limited availability of clinical samples and the exploratory design of the study, a formal power calculation was not feasible. Nevertheless, the sample size allowed for meaningful preliminary statistical evaluation of the parameters studied.

Asthma diagnoses were confirmed in accordance with the 2024 Global Initiative for Asthma guidelines [1]. Ethical approval for the above procedures was granted by the local ethics committee (TJ-IRB202408057).

### 2.2. Mouse Models

Transgenic *Hk2^flox/flox^* (strain number: T009288) and *Scgb1a1-iCre^+/+^* (strain number: T052678) mice were purchased from Gem Pharmatech (Nanjing, China). To generate mice with club cell-specific HK2 knockout, *Hk2^flox/flox^* mice were crossed with *Scgb1a1-iCre^+/+^* mice. Littermate *Hk2 ^flox/flox^ Scgb1a1-iCre^-/-^* mice, referred to as HK2-C mice, were used as controls. Genotyping of each mouse was performed using PCR (Appendix A).

All mice (8–12 weeks old) were housed in a specific-pathogen-free animal facility. Mice were randomly allocated to various groups, and the assessment was performed by an evaluator who was blinded to the treatment assignments. To induce asthma in mice, one group was sensitized with ovalbumin (OVA; 100 μg) and aluminum hydroxide (Alum; 100 μL) on days 0, 7, and 14 to induce eosinophilic airway inflammation. Intranasal sensitization with 10 μg OVA and 1 μg lipopolysaccharide (LPS; Sigma-Aldrich, St. Louis, MO, USA; prepared in saline) was given to another group on days 0, 1, 2, and 14. From days 21 to 23, both groups received daily intranasal OVA administration (1 mg for eosinophilic airway inflammation and 50 μg for eosinophilic airway inflammation. To investigate the effects of HK2 inhibition, mice were intraperitoneally injected with 2 g/kg 2-DG 12 h prior to each challenge. Control mice received an equivalent amount of saline or phosphate-buffered saline (PBS). For subsequent analysis, each animal was anesthetized with pentobarbital 24 h after the final exposure. All procedures involving animals were approved by the Animal Care and Use Committee at Tongji Hospital, Huazhong University of Science and Technology (TJH-202307031).

### 2.3. Airway Responsiveness to Methacholine

24 h after the final administration with saline or OVA, respiratory system resistance, compliance, and airway tissue elasticity were assessed using aerosolized methacholine (Sigma-Aldrich) via an invasive method with the flexiVent system (SCIREQ, Montreal, QC, Canada). Aerosolized methacholine was administered at 0–50 mg/mL, and the peak values of three indicators at each dose were recorded.

### 2.4. Antibodies and Reagents

HK2 (ProteinTech, Wuhan, China, 22029-1-AP, RRID: AB_11182717), PPIF (ProteinTech, Wuhan, China, 18466-1-AP, RRID: AB_2169273), VDAC1 (ProteinTech, Wuhan, China, 55259-1-AP, RRID: AB_10837225), Scgb1a1 (ProteinTech, Wuhan, China, 10490-1-AP,RRID: AB_2183285), β-actin (ProteinTech, Wuhan, China, 66009-1-Ig, RRID: AB_2687938), and IgG (ProteinTech, Wuhan, China, 98136-1-RR,RRID: AB_3672282), Poly (ADP-Ribose)-Polymerase (PARP) (Cell Signaling Technology, Danvers, MA, USA, #9542,RRID: P09874), Bax (Cell Signaling Technology, Danvers, MA, USA, #2772,RRID: AB_2909613), cleaved caspase-3 (Cell Signaling Technology, Danvers, MA, USA, #9661,RRID: AB_2341188), caspase-3 (Cell Signaling Technology, Danvers, MA, USA, #9662,RRID: AB_344892), and B-cell lymphoma 2 protein (Bcl-2) (Abcam, Cambridge, MA, USA, ab182858,RRID: AB_182858), recombinant IL-4 (MedChemExpress, Monmouth Junction, NJ, USA, HY-P70445), recombinant IL-13 (MedChemExpress, New Jersey, USA, HY-P70568), recombinant IL-6 (MedChemExpress, New Jersey, USA, HY-P7044), recombinant IL-8 (MedChemExpress, HY-P70569), recombinant IL-1β (MedChemExpress, HY-P7028), recombinant IL-33 (MedChemExpress, HY-P7041), LPS (Sigma-Aldrich, St.Louis, MO, USA), HDM (Greer Laboratories, Lenoir, NC, USA, Lot#421176), recombinant TNF-α (PeproTech, London, UK, 300-01A-50UG), DMEM/F12 (BasalMedia, Shanghai, China, L310KJ), Pronase E (Solarbio, Beijing, China, P8360), retinoic acid (MedChemExpress, New Jersey, USA, HY-14649), epidermal growth factor (MedChemExpress, New Jersey, USA, HY-P7076), bovine pituitary extract (Macgene, Beijing, China, CC023), insulin (Macgene, Beijing, China, CC101), transferrin (Macgene, Beijing, China, CC-109), cholera toxin (Macgene, Beijing, China, CC104).

### 2.5. Histopathology, Immunohistochemical and Immunofluorescence Analysis

Mouse left lung tissues and human tracheal biopsies were collected and fixed in fresh 4% neutral-buffered paraformaldehyde for 24 h at ambient temperature. The samples were embedded in paraffin and prepared for histological analysis. Hematoxylin and eosin staining (HE) was performed on mice lung sections. immunohistochemical staining of HK2 (anti-HK2, 1:500) was performed on human tracheal biopsy. Immunofluorescence co-staining of HK2 (anti-HK2, 1:500) and Scgb1a1 (anti-Scgb1a1, 1:400) were incubated on mice lung sections overnight.

### 2.6. Cell Culture and Treatment

Human bronchial epithelial cell line Beas-2B (purchased from ATCC, Manassas, VA, USA) was cultured in RPMI 1640 medium supplemented with 10% fetal bovine serum (Gibco, Grand Island, NY, USA). Cells were treated 48 hrs with HDM (50 μg/mL), IL-4 (20 μg/mL), IL-13 (20 μg/mL), IL-33 (20 μg/mL), IL-1β (20 or 10 ng/mL), IL-6 (10 μg/mL), IL-8 (10 μg/mL), TNF-γ (10 μg/mL), TGF-β (5 μg/mL), and LPS (1 μg/mL).

Mouse tracheobronchial epithelial cells (MTEC) were isolated from wild type mice as reported in previous literature [31,32]. Mice respiratory tracts were digested with protease E (1.5 mg/mL) overnight in 4 °C. The second day, the cell suspension was filtered, centrifuged, and cultured in epithelial growth medium (DMEM/F12, fetal bovine serum, 10%; retinoic acid, 0.1 μM; epidermal growth factor, 0.025 μg/mL; bovine pituitary extract, 0.06 mg/mL; insulin, 0.01 mg/mL; transferrin, 5 μg/mL; cholera toxin, 0.1 μg/mL) to enrich.

### 2.7. RNA Sequencing

RNA sequencing was conducted by Bioyi Biotechnology Co., Ltd. (Wuhan, China). Trizol Reagent (Invitrogen Life Technologies, Waltham, MA, USA) was utilized to extract total RNA. Following RNA quality and quantity assessment, libraries for sequencing were constructed and sequenced with the DNBSEQ-T7 platform in PE150 mode. Differential gene expression was analyzed using DESeq2 (v1.30.1) with the following criteria |log2FoldChange| > 1 and padj ≤ 0.05. Gene Ontology (GO) enrichment analysis of differentially expressed genes (DEGs) was conducted with the clusterProfiler package (v3.18.1), with the hypergeometric distribution method used to calculate *p*-values. GO terms with significantly enriched DEGs were identified at a threshold of padj ≤ 0.05 to reveal their primary biological functions. Additionally, Kyoto Encyclopedia of Genes and Genomes (KEGG) pathway enrichment analysis was performed using clusterProfiler (v3.18.1), to identify significantly enriched pathways (padj ≤ 0.05).

### 2.8. Western Blotting

Lung tissues and cultured cells were lysed with RIPA lysis buffer, and the extracted proteins were subjected to Western blotting. The proteins were transferred to PVDF membranes and incubated overnight with primary antibodies (anti-HK2, 1:5000; anti-β-Actin,1:5000; anti-PARP, 1:1000; anti-Bax,1:1000; anti-cleaved caspase-3,1:1000; anti-caspase-3, 1:2000; anti-Bcl-2, 1:2000; anti-PPIF, 1:4000; anti-VDAC1, 1:4000). Membrane visualization was achieved using the Gel Doc XR+ System (Bio-Rad, Hercules, CA, USA) following a previous study. Band intensities were quantified using Image J software (v.1.52a).

### 2.9. Flow Cytometry

The Annexin-V-FITC/PI Apoptosis Detection Kit (YEASEN, Shanghai, China) was utilized to detect apoptosis. Cell supernatant and cells were collected, washed with pre-cooled PBS twice, and then resuspended in 100 µL of binding buffer containing Annexin-V and PE (5 µL each). Cells were incubated at ambient temperature for 10–15 min in a light-proof environment. Fluorescence signals were analyzed by flow cytometry (Bida BioTech Co., Ltd., Changzhou, China).

### 2.10. Quantitative RT-PCR Analysis

Total RNA was extracted from cultured cells and mouse lung tissues using Trizol (Takara, Kusatsu, Shiga, Japan), which was subsequently reverse transcribed into cDNA. SYBR Premix Ex Taq (Takara, Kusatsu, Shiga, Japan) was used for quantitative RT-PCR. Quantitative RT-PCR was carried out using SYBR Premix Ex Taq (Takara, Kusatsu, Shiga, Japan). Relative mRNA expression was determined using the 2^−ΔΔCT^ method, with mouse β-actin or human 18S rRNA as endogenous controls. The target gene primers are listed in Appendix A.

### 2.11. ELISA

Bronchoalveolar lavage fluid (BALF) collected from model animals was subjected to centrifugation at 1000 rpm for 5 min. The concentrations of KC and CXCL20 in BALF were measured using ELISA kits following the manufacturer’s instructions purchased from Boster (Wuhan, China).

### 2.12. Oxygen Consumption and Glycolytic Rates

Oxygen consumption rate (OCR) and glycolytic rates were measured using a Seahorse XF24 Cellular Flux Analyzer (Agilent, CA, USA). In brief, Beas-2B cells (3000 cells/100 µL per well) were seeded into microplates and cultured in 37 °C with 5% CO₂ for 48 h prior to the assay. Following washing with non-buffered XF Base Medium RPMI 1640, the cells were placed in a CO₂-free/37 °C incubator for 1 hr before the assay. Selective inhibitors and assay medium were added to the respective wells. The final concentrations of inhibitors were as follows: rotenone/antimycin (0.5 μM), oligomycin (1.5 μM), carbonyl cyanide-4-(trifluoromethoxy) phenylhydrazone (1.5 μM), and 2-DG (50 mM).

### 2.13. Co-Immunoprecipitation (Co-IP) Assay

Co-IP was performed using the kit purchased from Cell Signaling Technology (Danvers, MA, USA, 70024s). Beas-2B cells were seeded in a 100-mm cell culture dish. When the cell confluency reached 30–40%, the cells were exposed to IL-1β (20 ng/mL) for 48 h. Then, cells were washed thrice with pre-chilled PBS and harvested. They were then harvested and lysed in NP-40 lysis buffer containing phosphatase/protease inhibitor cocktail (MedChemExpress). The lysates were subjected to centrifugation, and the supernatants were harvested and cleared with protein G magnetic beads (20 µL) for 2 h at 4 °C. The pre-cleared supernatants were then incubated overnight at 4 °C with rotation using anti-HK2 or anti-IgG to form immunocomplexes. The next day, the immunocomplex solutions were incubated with pre-cleared protein G magnetic beads (20 µL) at 4 °C for 3 h with rotation. Finally, the immunocomplexes were eluted from the protein G magnetic beads and subjected by western blotting.

### 2.14. Statistical Analysis

GraphPad Prism v7.0 software was used for statistical analysis, with comparisons made across at least three independent replications. Data were expressed as mean ± standard error of the mean. The statistical significance between two groups was assessed using Student’s *t*-test, while comparisons across multiple groups were analyzed with either one-way ANOVA or two-way ANOVA with a Tukey-Kramer test. A significance level of *p* < 0.05 was applied to all analyses with the following notation: *, *p* < 0.05; **, *p* < 0.01; ***, *p* < 0.001; ****, *p* < 0.0001.

## 3. Results

### 3.1. Altered Glucose Metabolism in Asthma

Non-targeted metabolomics analysis was performed on 24 asthma patients and 20 healthy control (HC) subjects. A total of 400 metabolites exhibited significant changes (|Fold change| ≥ 1) in asthma plasma compared to HCs. Pathway enrichment analysis reveals significant alterations in the glycolysis, tricarboxylic acid cycle (TCA), and lipid metabolism (Figure 1A). While our previous research has investigated disruptions in lipid metabolism [33], the present study focuses on glycolysis. Among glycolytic metabolites, both pyruvate and α-D-glucose were notably elevated in the plasma of asthma patients (Figure 1B,C). Notably, α-D-Glucose shows a strong correlation with clinical indicators related to asthma diagnosis and prognosis (Figure 1D,E). We further analyzed the levels of key TCA cycle and glycolysis enzymes in the PBMCs from asthma patients and mice. The expression levels of HK2, PFKM, PKM2, PDHB, PDHA1, PDK1, SLC2A1 were significantly upregulated in PBMCs of asthmatic patients (Figure 1F–M) and in asthmatic mice (Appendix A). Macrophages are critical to the immune system in asthma pathogenesis. Previous studies have suggested that M1 macrophage primarily relies on glycolysis as their main energy source [18,34]. We examined glycolytic enzyme levels in bone marrow-derived macrophages. In contrast to in vivo findings, the levels of glycolysis-related enzymes showed the opposite trend (Appendix A).

### 3.2. HK2 Is Upregulated in Asthma

Airway epithelium is a key structural component in the pathogenic process of asthma, yet the expression and functional role of HK2 in airway epithelial cells remain unexplored in asthma. Immunohistochemical analysis of airway biopsies from seven asthma patients and eight healthy controls confirmed a significant elevation in HK2 expression within the airway epithelial cells of asthma patients (Figure 2A). Similarly, HK2 expression was elevated in mice induced by OVA/Alum or OVA/LPS (Appendix A, Figure 2B,C). Co-immunostaining results further demonstrated that the elevated HK2 expression was localized primarily within the airway epithelium (Figure 2D). No significant changes in HK2 expression were detected when the cells were exposed to type 2 inflammatory cytokines (i.e., IL-13 and IL-4). A previous study has suggested that IL-1β is responsible for the enhanced glycolysis observed in asthmatic lungs, particularly in the airway epithelium [35]. Therefore, we hypothesize that IL-1β may be a key regulator of HK2 expression. To test this, we expanded our screening of inflammatory factors. The results revealed that IL-1β treatment led to a significant, dose-dependent increase in HK2 expression in Beas-2B cells (Figure 2E,F). In contrast, exposure to IL-6, IL-8, IL-33, TGF-β1, TNF-α, LPS, or HDM did not significantly alter HK2 expression (Figure 2E).

### 3.3. HK2 Deficiency Ameliorates Inflammation in Asthmatic Mice

To further investigate the function of HK2 in asthma, we generated *Hk2^flox/flox^-Scgb1a1-iCre^+/-^* mice, in which HK2 was specifically deleted in club cells (Appendix A). These mice were designated as the HK2-CKO group, while their littermate *Hk2^flox/flox^ Scgb1a1-iCre^−/−^* mice were used as controls (HK2-C mice). RT-PCR and Western blotting analyses confirmed a remarkable reduction in HK2 levels in the lung of HK2-CKO mice (Appendix A). The deficiency of HK2 in HK2-CKO mice was further confirmed by co-immunostaining of lung tissue (Appendix A). Additionally, primary mouse airway epithelium isolated from HK2-CKO also exhibited a marked loss of HK2 expression compared to HK2-C mice (Figure 3A). Considering HK2 is an important metabolic enzyme for growth, we assessed whether HK2 knockdown impacted overall growth. A four-week body weight assessment of the HK2-C and HK2-CKO mice demonstrated no significant difference between them (Appendix A).

As predicted, repeated OVA challenges for three days induced severe neutrophilic inflammation responses in HK2-C mice (Figure 3B), characterized by inflammatory cell infiltration to peribronchiolar connective tissues. On the contrary, HK2-CKO mice exhibited noticeably lower inflammatory (2.56 ± 0.08 vs. 1.64 ± 0.09, *p* < 0.0001; Figure 3C,D), indicating that airway epithelial-specific knockdown of HK2 markedly attenuated OVA-induced airway inflammation. A decreased inflammatory cell infiltration was also observed in the BALF (115.80 ± 7.29 vs. 84.17 ± 2.86, *p* < 0.0001; Figure 3E), particularly neutrophils (50.06 ± 5.68 vs. 25.00 ± 4.42, *p* < 0.01) and macrophages (32.11 ± 3.90 vs. 16.26 ± 1.65, *p* < 0.001) (Figure 3F). Furthermore, we further measured three AHR indicators: respiratory system resistance, compliance, and airway tissue elasticity. Upon normal conditions, HK2 deficiency had no significant impact on lung function. However, after methacholine stimulation in OVA-induced mice, both resistance and elasticity parameters were improved (Figure 3G).

Further analysis was conducted to assess cHK2′s influence on eosinophilic asthma (Appendix A). HK2 knockdown significantly ameliorated lung inflammation (2.16 ± 0.05 vs. 1.56 ± 0.07, *p* < 0.0001; Appendix A) and AHR (Appendix A) in OVA/Alum-induced mice. While the overall inflammatory cell infiltration in the BALF was reduced in HK2-C mice (135.00 ± 5.39 vs. 76.25 ± 4.99, *p* < 0.0001; Appendix A), a notable decline was observed in macrophages (28.57 ± 3.31 vs. 11.60 ± 1.79, *p* < 0.001; Appendix A). Collectively, these findings demonstrate that HK2 knockdown in airway epithelial cells protects against allergic airway inflammation and AHR induced by OVA.

### 3.4. HK2 Is Involved in Airway Epithelial Cell Apoptosis, Immune Response and Glycolysis

To elucidate the biological effects of HK2 on airway epithelial cells and the underlying mechanisms involved, we transfected Beas-2B cells with an HK2 plasmid. Western blot confirmed the overexpression of HK2 at a plasmid concentration of 0.5 µg (Appendix A). To explore HK2-induced transcriptomic changes, RNA sequencing was then performed on Beas-2B cells transfected with an HK2 plasmid following IL-1β exposure. There were 348 DEGs identified, with 128 downregulated and 210 upregulated genes (Figure 4A). GO analysis revealed that these DEGs were enriched in pathways related to chemokine activity, immune response and regulation of apoptosis (Figure 4B). Further Kyoto Encyclopedia of Genes and Genomes (KEGG) pathway analysis indicated significant involvement of the NF-kappa B signaling pathway, NOD-like receptor signaling pathway, and IL-17 signaling pathway (Figure 4C).

IL-1β was shown to dose-dependently induce apoptosis in Beas-2B cells, as featured by increased expression of cleaved Caspase-3, cleaved PARP1, and Bax, along with the upregulation of HK2 expression (Appendix A). Flow cytometry analysis showed that HK2 overexpression exacerbated the apoptotic rate of Beas-2B cells (Figure 4D). In addition, an increase in apoptosis marker expression was observed, with cleaved Caspase-3/Caspase-3 showing a statistically significant difference (Figure 4E). RT-PCR analysis was carried out to evaluate the transcription levels of IL-8, IL-6 and chemokine CCL20. All three cytokines remained highly expressed, with IL-8 exhibiting the highest expression levels (Figure 4F).

### 3.5. HK2 Knockdown Protects Airway Epithelial Cells Against Apoptosis and Inflammatory Cytokine Release

To further investigate the biological function of HK2, Beas-2B cells were transfected with HK2 siRNA. Among three HK2 siRNAs tested, siRNA-2# and siRNA-3# exhibited significant knockdown efficiency and were selected for further experiments (Appendix A). Both flow cytometry (Figure 5A) and Western blotting (Figure 5B) demonstrated that HK2 knockdown protected Beas-2B cells against apoptosis induced by IL-1β. Additionally, HK2 knockdown led to a decrease in Caspase-3 expression (Figure 5B). Moreover, RT-PCR results showed that HK2 knockdown significantly reduced the transcription levels of inflammatory cytokines, including CCL20, IL-8, and IL-6 (Figure 5C).

IL-1β treatment reduced mitochondrial respiration in Beas-2B cells, as reflected by a decrease in maximal oxygen consumption rate (the highest oxygen consumption rate cells can achieve), ATP production (the fraction of oxygen consumption allocated to ATP production by the mitochondria), and basal oxygen consumption rate (the energetic requirements of cells under normal conditions) (Figure 5D). It suggests that IL-1β enhances cellular activity, shifting metabolism toward glycolysis to meet increased energy demands. While HK2 knockdown had no significant impact on basal respiration or ATP production in unstimulated Beas-2B cells, it increased oxygen consumption and ATP production following IL-1β treatment (Figure 5D), implied the mitochondrial respiratory level partly converted. Glycolytic capacity was impaired due to HK2 deficiency, with no enhancement in glycolytic activity observed (Figure 5E).

### 3.6. HK2 Interacts with PPIF Protecting Beas-2b Cells Against Apoptosis in Asthma

Previous evidence has shown that HK2 mediates apoptosis by interacting with VDAC1 and PPIF, two key components of the mitochondrial permeability transition pore [36,37]. In line with this, HK2 overexpression led to an upregulation of both PPIF and VDAC1, though only PPIF exhibited a statistically significant increase (Figure 6A). Conversely, HK2 knockdown significantly reduced PPIF and VDAC1 expression (Figure 6B). Co-IP assay confirmed the direct interaction between HK2 and PPIF (Figure 6C), whereas no direct binding between HK2 and VDAC1 was detected (Figure 6D). Additionally, immunofluorescence analysis verified the co-localization of HK2 and PPIF in Beas-2B cells (Figure 6E). These findings suggest that HK2 upregulation contributes to epithelial apoptosis and airway inflammation in asthma through its interaction with PPIF.

### 3.7. HK2 Regulates Apoptosis and Immune Response In Vivo

To determine whether HK2 plays a similar functional role in vivo, we analyzed key apoptosis markers, including Bax/Bcl-2, cleaved PARP1/PARP1, and cleaved Caspase-3/Caspase-3. In HK2-C mice, these markers were significantly elevated in both types of airway inflammation, whereas in HK2-CKO mice, these markers were markedly reduced, with the exception of Bax/Bcl-2, which exhibited a downward trend but did not reach statistical significance (Figure 7A and Appendix A). Additionally, inflammatory cytokine levels were assessed. Following HK2 knockdown in the airway epithelium, cytokines such as IL-6, KC, IL-10, and chemokine CCL20 were significantly downregulated (Figure 7B,C). Similarly, Th2-related cytokines, including IL-13, IL-33, and chemokines CCL11 and CCL24, were significantly reduced (Appendix A). A significant elevation in IL-1β levels was found in mice induced by OVA/LPS and OVA/Alum, but the HK2 knockdown exerted no significant effect on IL-1β expression (Figure 7C and Appendix A). Furthermore, both PPIF and VDAC1 expression were downregulated in vivo following HK2 deletion, with PPIF showing statistically significant changes (Figure 7D and Appendix A). These findings indicate that HK2 regulates both airway epithelial apoptosis and immune response in vivo.

### 3.8. Inhibiting HK2 Activity by 2-DG Protected Mice from Airway Inflammation

To better evaluate the therapeutic potential of HK2 inhibition, we administered 2-DG, a glucose analog targeting hexokinases and impedes glycolysis, to asthmatic mice. Mice were given an intraperitoneal injection of 2-DG 12 h prior to each challenge, and all mice were euthanized 24 h after the final challenge. Hematoxylin and eosin staining and inflammation scoring revealed that 2-DG had no significant impact on airway inflammation scores under normal conditions. However, OVA-induced airway inflammation was markedly reduced following 2-DG treatment (Figure 8A–D). In alignment with previous findings, AHR was also ameliorated following HK2 inhibition (Figure 8E). Furthermore, inflammatory cytokines, including IL-4, IL-13, IL-33, and the chemokine CCL20, were significantly reduced by 2-DG treatment (Figure 8F). These results suggest that HK2 represents a promising therapeutic target for asthma treatment.

## 4. Discussion

Using non-targeted metabolomics analysis, we identified metabolic reprogramming in asthma patients, particularly in lipid and glucose metabolism. While our previous work emphasized lipid metabolism [33], this study highlights the role of glycolysis in asthma pathogenesis. Our findings demonstrate that HK2, a key enzyme in glycolysis, is overexpressed in the airway epithelium of asthma patients and mouse asthma models. IL-1β exposure induced HK2 expression in Beas-2B cells in a dose-dependent manner. In addition, HK2 deficiency in airway epithelial cells conferred protection to mice against airway inflammation and AHR. Notably, HK2 regulates immune responses, apoptosis, and glycolysis through its interaction with PPIF. Moreover, pharmacological inhibition of HK2 with 2-DG significantly improved AHR and pulmonary inflammation in asthmatic mice, identifying HK2 as a promising therapeutic target for managing asthma.

Asthma pathogenesis is influenced by metabolic reprogramming, which alters immune cell activation, function, and differentiation through mechanisms such as amino acid metabolism, glycolysis, and fatty acid oxidation. [15,38,39]. Our non-targeted metabolomics analysis of 24 asthma patients and 20 HCs revealed dysregulated lipid metabolism, TCA cycle, and glycolysis. Based on these findings, we previously identified the involvement of Sphingosine-1-Phosphate Receptor 4 in asthma, where it suppresses proinflammatory macrophage activation via JNK phosphorylation and formyl peptide receptor 2 [33]. Clinical studies have reported elevated lactate concentrations in the sputum [35] and serum [40,41] of asthma patients, correlating with lung function. In line with these findings, we observed increased levels of glycolytic metabolites including α-D-Glucose and pyruvate, in asthma patients. α-D-Glucose exhibited a strong correlation with clinical indicators of asthma, further highlighting the essential role of glycolysis in asthma.

Previous studies have demonstrated that HK2 is involved in multiple inflammatory and immune-related diseases through diverse mechanisms. In rheumatoid arthritis, HK2 promotes fibroblast-like synoviocyte activation and synovial hypertrophy, contributing to disease progression [25,26,27]. Similarly, in intestinal inflammation, elevated HK2 expression in epithelial cells induces cell death and mitochondrial dysfunction, while the gut microbial metabolite butyrate downregulates HK2, exerting protective effects against colitis [24]. In pulmonary fibrosis, TGF-β–induced HK2 expression has been shown to promote fibrotic remodeling, indicating its potential as a therapeutic target in fibrotic lung diseases [42]. Dynamin-related protein 1 regulates HK2, promoting LPS-induced glycolysis and proliferation in airway smooth muscle cells [43]. Although T cells are highly metabolically active, HK2 appears dispensable for T cell function, as HK2 deficiency does not impair T cell responses [28,44]. In the present study, we observed significantly elevated HK2 expression in the airway epithelium of asthma patients, suggesting that HK2 participates in asthma pathogenesis by contributing to airway epithelial cell apoptosis and inflammation. These findings expand the current understanding of HK2 beyond its classical metabolic role, highlighting its potential involvement in the complex immune-metabolic networks of asthma.

Airway epithelium is line of defense between the body and environmental allergens, pollutants, and pathogens [45,46]. Epithelial cells are metabolically active, requiring energy to maintain barrier integrity, mediate immune responses, and repair tissue damage [47,48]. Aberrant glycolysis in epithelial cells, marked by elevated lactate production, has been observed in asthma patients [35] and in mouse tracheal epithelial cells [19]. Unlike classical type 2 inflammatory cytokines, which did not alter HK2 expression, IL-1β signaling played a pivotal role in HK2 upregulation in asthma [35]. This aligns with our study, showing that HK2 upregulation in the airway epithelium is closely related to the IL-1β signaling. Given global HK2 deletion is embryonically lethal, we generate *Hk2 ^flox/flox^ Scgb1a1-iCre^+/^^-^* mice to explore how HK2 functions in asthma. Despite slightly slower ponderal growth, HK2-CKO mice showed no significant difference in lung pathology or function in comparison to control mice. However, following OVA/Alum or OVA/LPS exposure, the HK2-CKO group exhibited substantial protection against inflammation. Furthermore, treatment with 2-DG significantly ameliorated airway inflammation and AHR, supporting the involvement of HK2 in asthma pathogenesis.

Apoptosis plays a pivotal role in asthma pathogenesis [49,50]. In healthy airways, apoptosis is important in the airway cell cycle, allowing for the removal of damaged cells and regeneration of healthy cells [11,49]. However, in the asthmatic airway, this balance is disrupted due to impaired epithelial repair mechanisms [49,51]. Excessive and dysregulated apoptosis in asthma, driven by multiple factors such as pro-inflammatory cytokines, oxidative stress, and viral infections [11,15,52], results in disruption of the epithelial barrier, making harmful particles more easily to penetrate deeper into the airways, thereby triggering or exacerbating immune responses and increasing inflammation [50,52]. HK2 has been recognized for its role in apoptosis in several cancers [22,53,54]. Our sequencing analysis and experiments findings confirmed that elevated HK2 promotes epithelial apoptosis in asthma. While HK2 overexpression only partially exacerbated apoptosis in vitro, HK2 knockdown markedly reduced apoptosis both in vivo and in vitro.

HK2 is localized to two primary cellular compartments: cytosol and mitochondria [55]. Mitochondria-bound HK2 maintains mitochondrial membrane integrity and regulates the apoptotic process. VDAC1 prevents the secretion of cytochrome c from mitochondria and subsequent apoptotic cell death following Bax activation by interacting with HK2 [56]. PPIF plays a key role in regulating cell apoptosis by binding to HK2 by its PPIase activity [36]. In our studies, we found HK2 facilitated PPIF expression in vivo *and* in vitro, but VDAC1 remains largely unchanged. Although these findings support a functional link between HK2 and PPIF in regulating epithelial apoptosis, the precise integration of this pathway with other known regulators of epithelial cell death and airway inflammation in asthma remains unclear. Furthermore, given the heterogeneity of asthma endotypes, including type 2 and non-type 2 inflammation, it is plausible that the HK2-PPIF axis may differentially contribute to epithelial dysfunction across distinct asthma subtypes. Further studies are warranted to elucidate how this metabolic-apoptotic pathway interacts with broader immune-inflammatory networks involved in asthma pathogenesis.

The limitations of this study should also be mentioned. First, the number of asthma patients included in the sample was limited. Besides the lack of detailed assessment of atopic sensitization, as only total serum IgE levels were measured, and neither specific IgE nor skin prick testing was conducted. This may restrict the ability to fully evaluate the association between HK2 expression and specific asthma phenotypes or endotypes. Future research should expand the patient cohort and incorporate asthma subtyping to assess HK2 expression across different asthma phenotypes. Second, this study primarily focused on airway epithelial cells, while asthma is a complex disease involving both adaptive and innate immune responses. The elevated HK2 expression observed in airway epithelium may reflect downstream effects of the inflammatory microenvironment. The potential regulation of HK2 by immune-mediated pathways and its role in other immune cell populations remain unclear and warrant further investigation. Finally, the integration of HK2-PPIF signaling with other regulatory pathways was not fully explored in this study and warrants further investigation.

## 5. Conclusions

In summary, our study demonstrated that asthma patients exhibit multiple metabolic reprogramming, including in lipid and glucose metabolism. Elevated HK2 expression in airway epithelial cells regulates glycolysis, airway inflammation, and apoptosis, thereby contributing to airway inflammation and AHR in asthma. Mechanistically, HK2 interacted with PPIF to regulate epithelial apoptosis. These findings support the hypothesis that HK2 may serve as a promising therapeutic target for asthma treatment.

## Figures and Tables

**Figure 1 cells-14-01004-f001:**
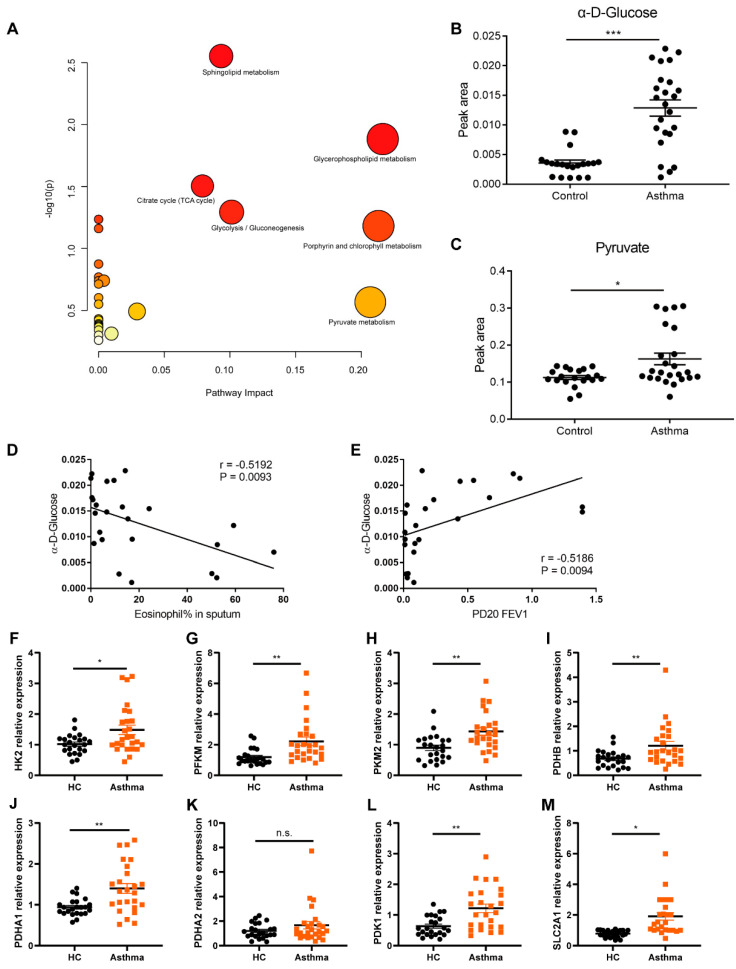
Altered glucose metabolism in asthma. (**A**) Non-targeted metabolomics pathway enrichment analysis in asthma patients. (**B**,**C**) Level of α-D-glucose (**B**) and pyruvate (**C**) in the plasma of asthma. (**D**,**E**) The correlation between plasma α-D-glucose and PD20 FEV1 (**D**) as well as the proportion of eosinophils in sputum (**E**). (**F**–**M**) RT-PCR analysis of the key enzymes expression involved in glucose metabolism: HK2 (**F**), PFKM (**G**), PKM2 (**H**), PDHB (**I**), PDHA1 (**J**), PDHA2 (**K**), PDK1 (**L**), and SLC2A1 (**M**) in PBMCs from asthma patients and healthy controls. Each group included 20–24 samples. HC = healthy controls, PBMCs = peripheral blood mononuclear cells, HK2 = hexokinase 2, PFKM = phosphofructokinase, muscle type, PKM2 = pyruvate kinase M2, PDHB = pyruvate dehydrogenase E1 subunit beta, PDHA1 = pyruvate dehydrogenase E1 subunit alpha 1, PDHA2 = pyruvate dehydrogenase E1 subunit alpha 2, PDK1 = pyruvate dehydrogenase kinase 1, SLC2A1 = solute carrier family 2 member 1. The data were shown as mean ± SEM, statistical differences were determined with a student’s *t* test for comparison between the two groups. * *p* < 0.05, ** *p* < 0.01, *** *p* < 0.001, ns = not significant.

**Figure 2 cells-14-01004-f002:**
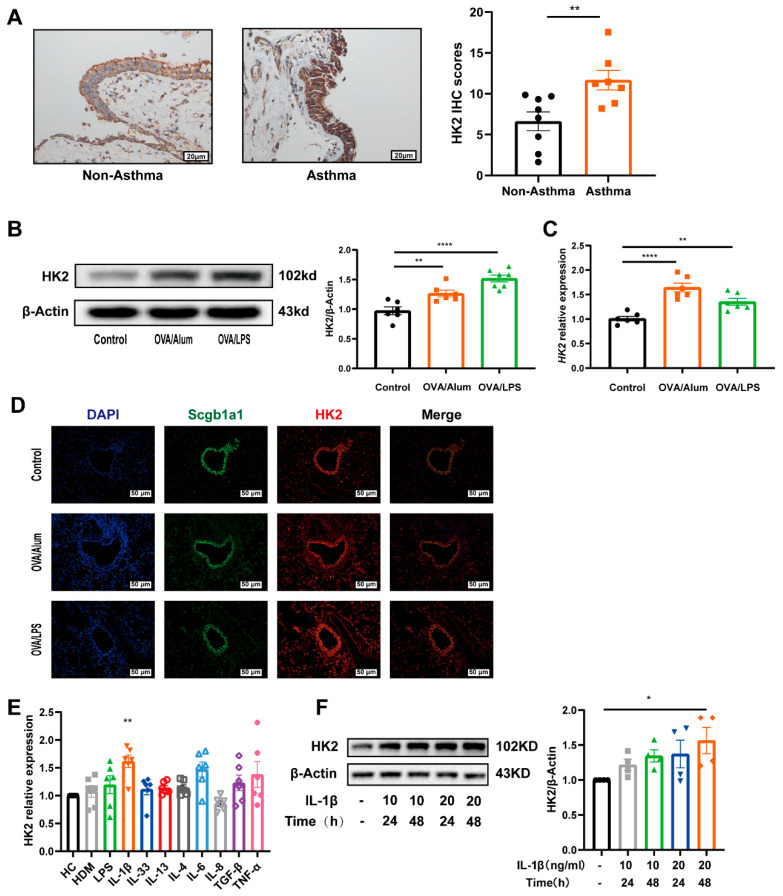
HK2 is upregulated in asthma. (**A**) Representative images showing HK2 protein immunostaining (brown color) in patients with asthma (*n* = 7) compared with that seen in healthy subjects (*n* = 8), HK2 protein relative expression was analyzed by Image-pro plus 6.0 and presented graphically. The images were captured under an original magnification of ×400. Scale bar, 100 μm. (**B**) Western blot analysis of HK2 in the lungs of experimental animals (*n* = 6 per group). (**C**) RT-PCR analysis of HK2 in the lungs of experimental animals (*n* = 6 per group). (**D**) Representative results of co-immunostaining of Scgb1a1 (secretoglobin family 1A member 1) and HK2 in the lung sections of experimental animals. The images were captured under an original magnification of ×200. Scale bar, 50 μm. (**E**) HK2 mRNA levels were measured after 48 h of stimulation with HDM, IL-1β, IL-33, IL-6, IL-8, LPS, TGF-β, or TNF-α (*n* = 6 per group). (**F**) Western blot results demonstrating HK2 expression in Beas-2B cells following treatment with IL-1β (*n* = 4 per group). Data are presented as mean ± SEM, Statistical differences were determined with a student’s t test for comparison between the two groups and one-way ANOVA with a Tukey-Kramer test for multiple-group comparisons. * *p* < 0.05, ** *p* < 0.01, **** *p* < 0.0001.

**Figure 3 cells-14-01004-f003:**
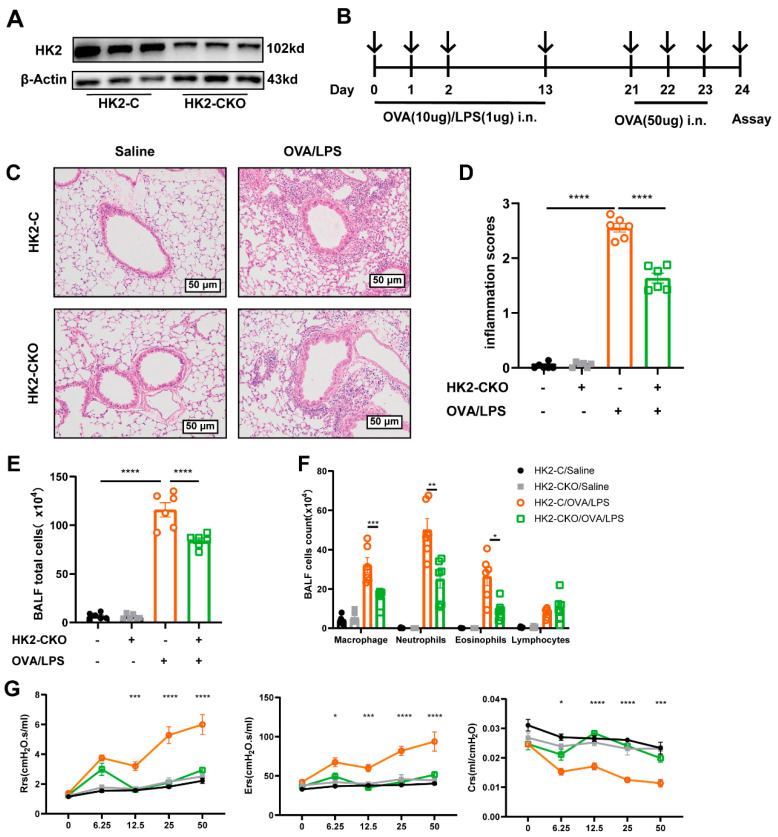
HK2 deficiency suppresses inflammation in asthmatic mice. (**A**) HK2 expression in the primary mouse tracheal epithelium extracted from HK2 conditional knockout (HK2-CKO) mice and HK2 control (HK2-C) mice. (**B**) Experimental scheme outlining the procedures. (**C**,**D**) Representative images and statistical graphs (*n* = 6) of lung histology from HK2-CKO and HK2-C mice following OVA/LPS. (**E**) Total cell counts in bronchoalveolar lavage fluid (BALF). (**F**) Differential counts of inflammatory cells in the BALF from HK2-CKO and HK2-C mice following OVA/LPS induction (*n* = 6). (**G**) Airway hyperresponsiveness (AHR), including respiratory system resistance (Rrs), elastance (Ers), and compliance (Crs), measured 24 h following the final challenge using flexiVent (*n* = 6). H&E = hematoxylin and eosin; Rrs = respiratory system resistance; Ers = respiratory system elastance; Crs = respiratory system compliance. The data were shown as mean ± SEM, statistical differences were determined with a student’s t test for comparison between the two groups and one-way ANOVA or two-way ANOVA with a Tukey-Kramer test for multiple-group comparisons. * *p* < 0.05, ** *p* < 0.01, *** *p* < 0.001, **** *p* < 0.0001.

**Figure 4 cells-14-01004-f004:**
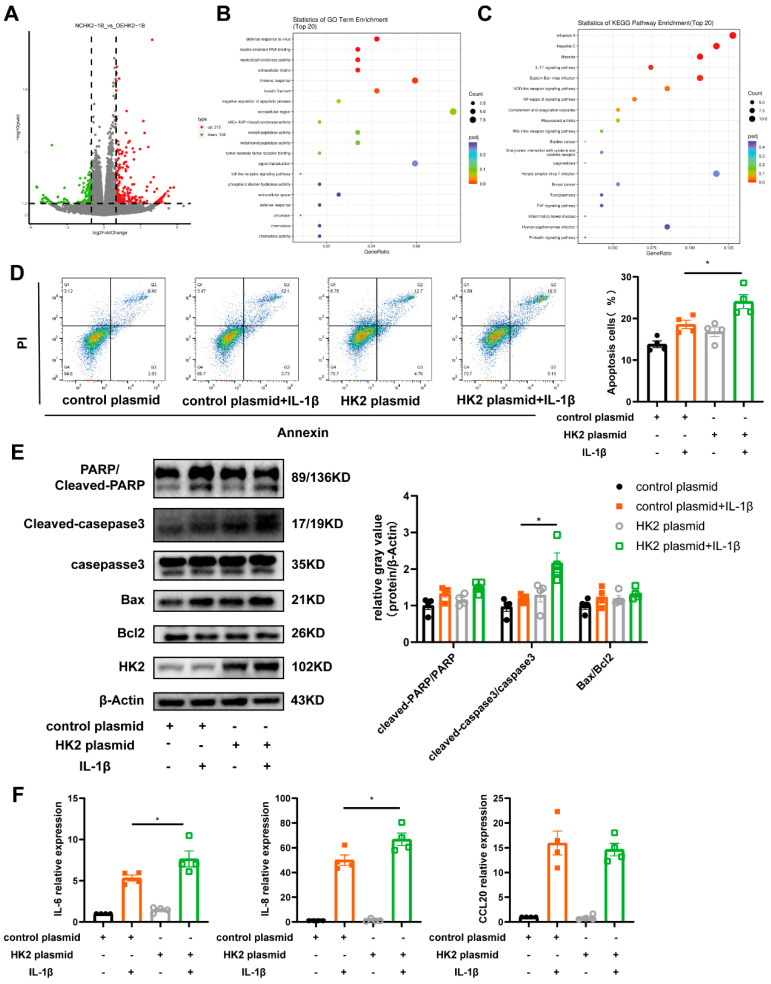
HK2 participates in airway epithelial apoptosis, immune response and glycolysis through multiple signaling pathways. (**A**) Volcano plot showing transcriptome changes from Beas-2b cells transfected with HK2 plasmid or control plasmid. Fold change ≥1.5. (**B**,**C**) Enriched GO terms (**B**) and KEGG pathways (**C**) analysis of DEGs. (**D**) Flow cytometry analysis of the apoptosis of Beas-2b cells transfected with plasmid and treated with IL-1β for 48 h (*n* = 4 per group). (**E**) Western blot analysis of HK2, PARP1, cleaved PARP1, caspase-3, cleaved caspase-3, Bax, and Bcl-2 in Beas-2b cells transfected with plasmid and treated with IL-1β for 48 h (*n* = 4 per group). (**F**) Inflammatory mediators. Data are presented as mean ± SEM, Statistical differences were determined with one-way ANOVA with a Tukey-Kramer test for multiple-group comparisons. * *p* < 0.05.

**Figure 5 cells-14-01004-f005:**
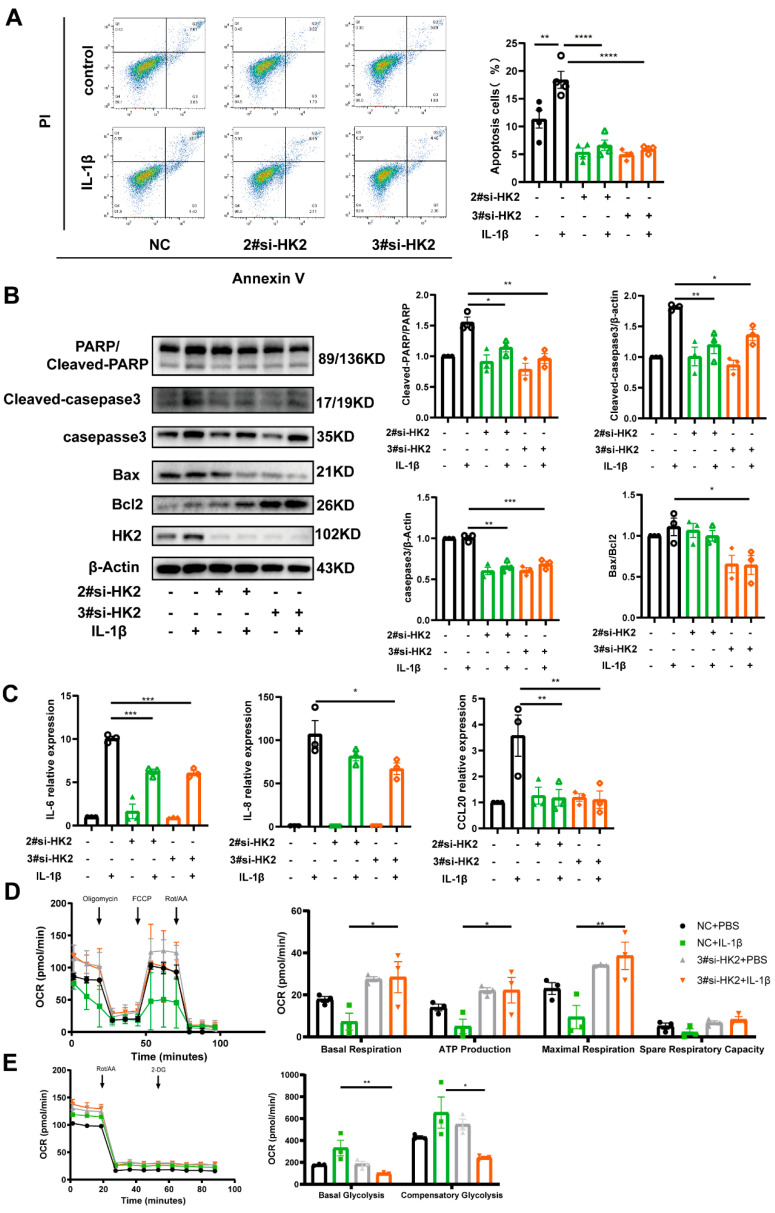
Ablation of HK2 protects Beas-2b cells from apoptosis and inflammatory reaction. (**A**) Flow cytometry analysis of the apoptosis of Beas-2b cells transfected with si-RNA and treated with IL-1β for 48 h (*n* = 4 per group). (**B**) Western blot analysis of HK2, PARP1, cleaved PARP1, caspase-3, cleaved caspase-3, Bax, and Bcl-2 protein expression in Beas-2b cells following transfection with si-RNA and treated with IL-1β for 48 h (*n* = 3 per group). (**C**) Inflammatory mediators. (**D**) Basal, maximal and ATP-producing oxygen consumption rates (OCR) were measured by Seahorse after oligomycin, FCCP, and rotenone/antimycin A (Rot/AA) treatments in silenced HK2 Beas-2b cells. (**E**) Basal and compensatory glycolysis were measured by Seahorse after Rot/AA and 2-DG treatments in silenced HK2 Beas-2b cells. OCR = oxygen consumptionrates, FCCP = carbonyl cyanide-p-trifluoromethoxyphenylhydrazone, Rot/AA = rotenone/antimycin A. Data are presented as mean ± SEM, Statistical differences were determined with one-way ANOVA with a Tukey-Kramer test for multiple-group comparisons. * *p* < 0.05, ** *p* < 0.01, *** *p* < 0.001, **** *p* < 0.0001.

**Figure 6 cells-14-01004-f006:**
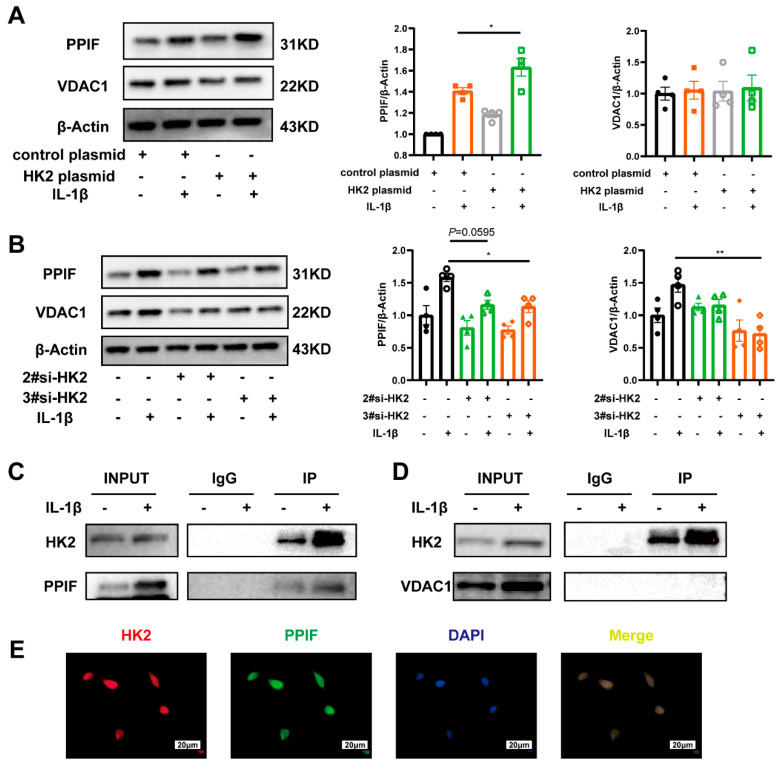
HK2 interacts with VDAC1 and PPIF in asthma. (**A**) Western blot analysis of VDAC1 and PPIF protein expression in Beas-2b cells following transfection with plasmid and treated with IL-1β for 48 h (*n* = 4 per group). (**B**) Western blot analysis of VDAC1 and PPIF protein expression in Beas-2b cells following transfection with si-RNA and treated with IL-1β for 48 h (*n* = 4 per group). (**C**) Co-immunoprecipitation analysis of HK2 and PPIF. (**D**) Co-immunoprecipitation analysis of HK2 and VDAC1. (**E**) Representative fluorescence microscopy images of HK2 (Cy3, red), PPIF (FITC, green), and DAPI (blue), the scale bar 20 um. Data are presented as mean ± SEM, Statistical differences were determined with one-way ANOVA with a Tukey-Kramer test for multiple-group comparisons. * *p* < 0.05, ** *p* < 0.01.

**Figure 7 cells-14-01004-f007:**
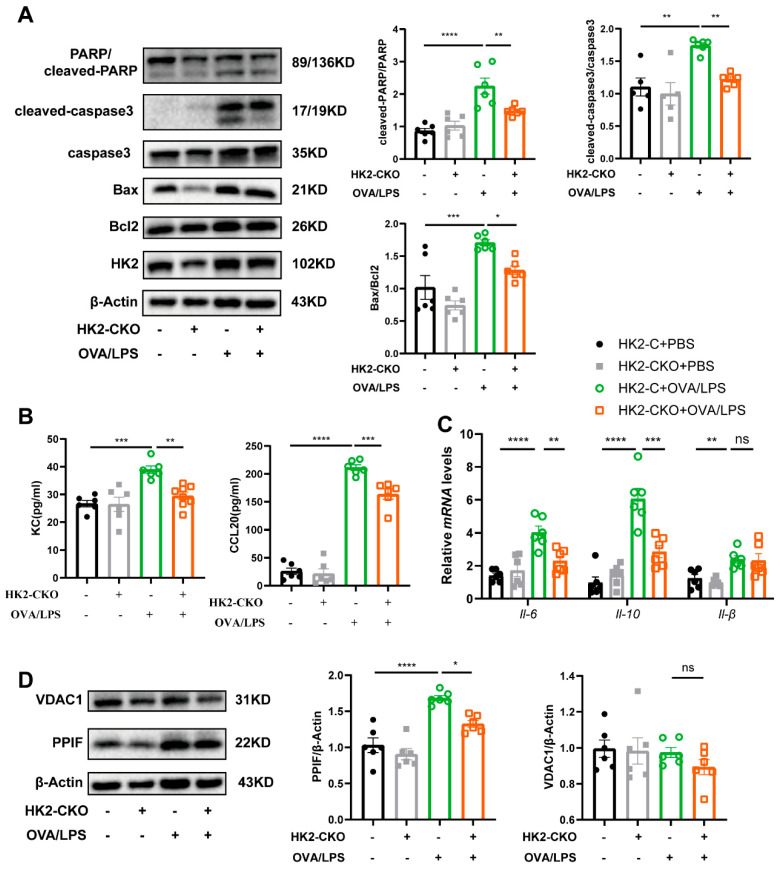
HK2 regulates apoptosis and immune response in vivo. (**A**) Western blot analysis of HK2, PARP1, cleaved PARP1, caspase-3, cleaved caspase-3, Bax, and Bcl-2 in asthmatic mice induced by OVA/LPS (*n* = 6 per group). (**B**) Protein levels of KC and CCL20 in BALF supernatants of OVA/LPS-immunized HK2-C and HK2-CKO mice (*n* = 6 per group). (**C**) mRNA levels of IL-6 and IL-10 in lungs of OVA/LPS-immunized HK2-C and HK2-CKO mice (*n* = 6 per group). (**D**) Western blot analysis of VDAC1 and PPIF protein expression in vivo. Data are presented as mean ± SEM, Statistical differences were determined with one-way ANOVA with a Tukey-Kramer test for multiple-group comparisons. * *p* < 0.05, ** *p* < 0.01, *** *p* < 0.001, **** *p* < 0.0001, ns = not significant.

**Figure 8 cells-14-01004-f008:**
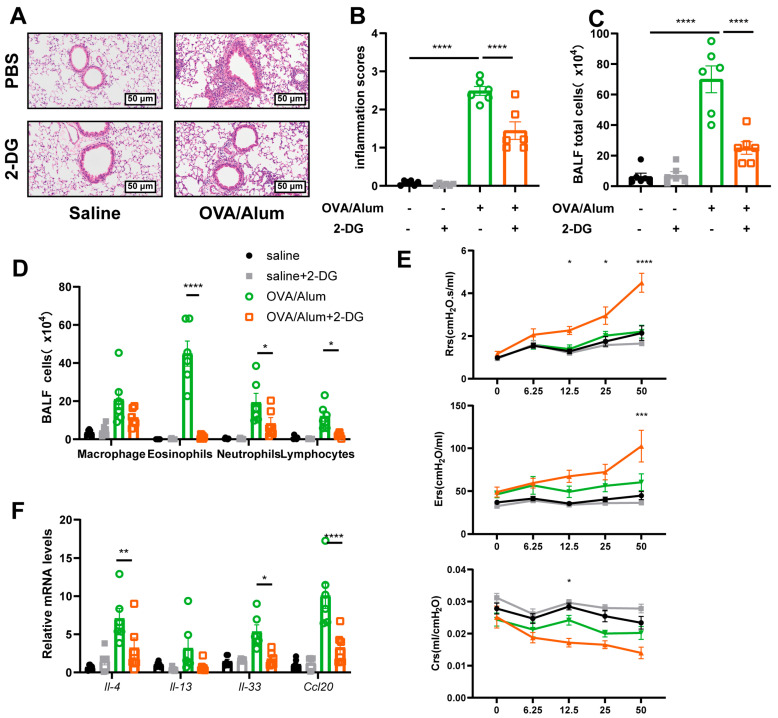
Inhibiting HK2 activity by 2-DG protected mice from airway inflammation. (**A**,**B**) Images and statistical graph (*n* = 6) of lung histology. (**C**) BALF total cells counts. (**D**) Differential counts of inflammatory cells in the BALF (*n* = 6). (**E**) mRNA levels of IL-4, IL-13, IL-33 and CCL20 in lungs of OVA-immunized mice treated with 2-DG or not (*n* = 6 per group). (**F**) AHR, including Rrs, Errs, and Crs, was recorded 24 h after the last flexiVent challenge. (*n* = 6). Elevated HK2 expression in airway epithelial cells regulates airway epithelial cells glycolysis, airway inflammation and cell death, contributing to the pathological of asthma. AHR = airway hyperresponsiveness, Rrs = respiratory system resistance; Ers = respiratory system elastance; Crs = respiratory system compliance. The data were shown as mean ± SEM, statistical differences were determined with one-way ANOVA or two-way ANOVA with a Tukey-Kramer test for comparison. * *p* < 0.05, ** *p* < 0.01, *** *p* < 0.001, **** *p* < 0.0001.

## Data Availability

Data supporting reported results can be found at Tian, Zhen (2025), “Upregulated Hexokinase-2 in Airway Epithelium Regulates Apoptosis and Drives Inflammation in Asthma via peptidylprolyl isomerase F”, Mendeley Data, V3, doi: 10.17632/7z76yw849w.3.

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
