# Peer review of "Upregulated Hexokinase-2 in Airway Epithelium Regulates Apoptosis and Drives Inflammation in Asthma via Peptidylprolyl Isomerase F"

_cells, 2025, doi:10.3390/cells14131004_

Round 1

Reviewer 1 Report

Comments and Suggestions for Authors

Reduction in HK2 levels in the lung of HK2-CKO mice would be expected. Also, attenuation of OVA-induced airway inflammation is the consequence. 

The experiments were properly performed. However, there are problems with the causality - see below.

It is not true that (lines 450-451) 'Metabolic reprogramming, amino acid metabolism glycolysis, and fatty acid oxidation is implicated in asthma pathogenesis by mediating immune cell activation, function, and differentiation).' In fact, it is opposite - immune cell activation, function, and differentiation is the cause for increased amino acid metabolism glycolysis, and fatty acid oxidation.

Similarly (lines 476-467) 'Elevated HK2 475 has been implicated in inflammatory diseases and autoimmune diseases. HK2 contributes to the progression of rheumatoid arthritis by regulating fibroblast-like synoviocyte activation and synovial hypertrophy. This is completely opposite.

As well, the next sentence 'Upregulated HK2 in the intestinal epithelium regulates epithelial cell death and mitochondrial dysfunction, while gut microbial metabolite butyrate downregulates HK2, thereby protecting against colitis.'

Please look carefully for the sequence of pathways!

Line 503 - Apoptosis, or programmed cell death - is any difference between them?

Line 529 'The potential function of HK2 in other immune cell populations remains unclear and warrants further exploration.' A consequence of a previous misunderstanding.

Miore:

 line 56 - glioblastoma is not cancer - just glioblastoma

line 56 - acute kidney injury is not an inflammatory disease (just remove)

Description of Figure 2 has to be improved ('Asthma expresses more HK2' is not acceptable)

Line 523  First, First, 

Reviewer 2 Report

Comments and Suggestions for Authors

The authors submit an interesting paper concerning the study of the role of hexokinase 2 in the pathogenesis of asthma.

They carry out this experimental work in the wake of their previous work on lipid metabolism in this pathological condition. In order to investigate the role of glycolysis in this context, they focus on a key enzyme of this metabolic pathway, namely hexokinase, which has recently been the focus of numerous studies, as its specific mutations are correlated with the development of certain pathologies, including tumour forms.

The authors have well organised the paper. The abstract is sufficiently informative and the introduction is comprehensive to indicate the field in which the aims of the study are framed. The bibliographical references are adequate and up-to-date. The materials and methods section is comprehensive, well organised to report what was done in a complex study that involved analysis of human samples, in the laboratory animal and in cell cultures.

The Authors carried out the experimental work in a rigorous and thorough manner, they carried out the experiments using histological, immunological, cytofluorimetric, biochemical and molecular biology techniques in a correct manner, always incorporating the appropriate controls in each type of investigation they carried out. The critical spirit with which they approached the experimentation is also reflected in the statistical analysis of the experimental results, which was carried out correctly, giving the correct weight and meaning to the results obtained.

The results of the study are also well reported. The design of the study was done in a manner consonant with the objectives the authors had set themselves and this is seen in the linearity of the conduct of the experimental investigations. Each result is well explained and commented on in its biological significance, and the images, figures and tables (also in the supplementary materials) are informative and greatly help the reader, who must be clear about each step in the study in order to understand and appreciate the next step in the experiments.

On the basis of their study, the central role of an enzyme such as hexokinase 2 in asthmatic pathology becomes clear. Indeed, with their work, the authors manage to provide interesting and clear information about the involvement of hexokinase 2 in the processes of inflammation and apoptosis underlying this widespread pathological condition, information that will form the basis of inevitable subsequent studies to clarify an extremely complex picture linking normal cell metabolism with a pathological condition.

On the strength of their previous work on lipid metabolism, the authors well valorise the results of their work, pointing to hexokinase 2 as a possible pharmacological target, but at the same time noting the critical issues and limitations of their study. Hexokinase is an enzyme with a basic role in the glucose metabolism of the cell and therefore of the organism, and consequently great care must be taken in modulating its activity pharmacologically.

The Authors have worked well and very hard, their paper is a good paper

Reviewer 3 Report

Comments and Suggestions for Authors

Upregulated Hexokinase-2 in Airway Epithelium Regulates   Apoptosis and Drives Inflammation in Asthma via PPIF

The topic is interesting. However, some points need to be addressed.

Abstract

  • The sample sizes and characteristics of the human cohorts under study are not specified in the abstract. By including this information, possible issues about statistical power would be addressed and the findings' generalizability would be contextualized.

  • Although the abstract could be strengthened by briefly noting the preclinical evidence supporting this claim (e.g., the efficacy of 2-DG in murine models) and its possible limitations or next steps, the statement "targeting HK2 represents a promising therapeutic strategy" is appropriate.

  • Language & Grammar: Although the abstract is usually well written, little grammar changes might improve reading (e.g., "hexokinase catalyzes the initial rate-limiting step in glucose metabolism by transforming glucose into glucose-6-phosphate").
  • Good practice is defining all abbreviations (e.g., PPIF, VDAC1, 2-DG).

Introduction

  • The introduction talks on the connection between metabolism and asthma, but it might also touch on recent developments in immunometabolism and how focusing metabolic pathways is becoming a therapeutic approach in inflammatory disorders. This would highlight still more the translational relevance of the work.

  • Little linguistic changes could improve clarity (e.g., rework difficult sentences for concision).

Materials and methods

  • Although the total sample sizes are appropriate, the relative small number of human airway biopsy samples (asthma n=7, controls n=8) could restrict statistical power and generalizability. Although this restriction is admitted in the conversation, it could also be addressed here very briefly.
  • Neither whether animals or samples were randomized nor whether researchers were blinded to group allocation throughout data collecting and analysis, the section leaves open questions. Including this material would help to lower possible bias and increase methodological rigor even more.
  • Although most protocols are sufficiently detailed, some (e.g., exact conditions for primary mouse epithelial cell growth, specifics of in-house quality control for metabolomics) may profit from further explanation or references to accepted protocols.

Results

  • Although the main conclusions are adequately stated, the results part would benefit from more frequent inclusion of particular quantitative values (e.g., fold changes, p-values) in the text to complement the figures and improve reader comprehension.
  • Although the main conclusions are adequately stated, the Results section would benefit from more regular inclusion of particular quantitative data (e.g., fold changes, p-values) in the text to supplement the figures and improve reader comprehension.
  • There are just seven asthma cases and eight controls in the human sample size for airway biopsies; asthma subtypes are not differentiated. Although this is a drawback of the study overall, stating it in the Results will help to reduce overgeneralization.

Discussion

  • Although the relationship between HK2 and PPIF is underlined, the debate could go on on how this route interacts with other known regulators of epithelial death and inflammation in asthma, and whether it may interact with existing asthma endotypes or phenotypes.
  • Recent studies have linked severe or T2-low asthma to mitochondrial dysfunction in airway epithelial cells. Combining these ideas might help to underline the significance of metabolic pathways in several kinds of asthma.
  • Though a more succinct summary of the main take-home messages at the end would increase readability, the debate is generally clear and well written.
  • Minimizing some repetition of past findings would help to concentrate the debate more precisely on interpretation and consequences.

Reviewer 4 Report

Comments and Suggestions for Authors

The manuscript entilted:Upregulated Hexokinase-2 in Airway Epithelium Regulates 2
Apoptosis and Drives Inflammation in Asthma via PPIF is generally well written and nicely presented

Some points can be considered to improve the quality of the present manuscript

1- Avoid using abbreviations in the title, and all  abbreviations should be defined 

2- Regarding the clinical study: The authors should state basis for selection of sample size 

3- details of western blot and immunohistochemistry should be stated in details including the dilutions used and time of incubation of antibodies

4-Could the authors improve the quality of resolution of fig 2 A  and 3C

5- Discussion needs further work as it is only a repetition of main findings ignoring the link between major pathways.

Reviewer 5 Report

Comments and Suggestions for Authors

With interest, I read the manuscript cells-3671480. The topic of this work is highly intriguing and research questions are well formulated. The experimental plan includes in investigations both in human and in mice. Several modern, up-to-date methods were applied. The results are in general interesting. The data are generally well interpreted and conclusions are drawn correctly.

I have several experimental design-related comments, and sometimes I am referring to the way the data are reported or how they should be discussed.

Comments (no special order):

  1. Please, highlight some more the role of airway epithelium in asthma, especially in the context of varying mechanisms underlying different forms of the disease, so-called endotypes (PMID: 31904412). It is important as it has some consequences for the assessment of the design and the results of the study. I do not mean here only phenotypes of asthma combined with obesity but general classification of asthma as type 2 or non-type 2 (with several other/overlapping names of those major type groups).
  2. You state that in your animal experiments, various asthma phenotypes were tested. First remark, please, do not write “asthma” with regard to asthma. What you do in mice is a “murine model of allergic airway inflammation mimicking human asthma”.
  3. Second, what types of asthma you mimic? OVA/Alum -> eosinophilic? OVA/LPS -> neutrophilic? It is never clearly stated, if I did not overlook anything. It would be fine, as various types of asthma can be mimicked in murine models (PMID: 30267575, 36271366).
  4. Further, from what I see, OVA/Alum model (presumably eosinophilic, type 2, etc.) was used only in a few experiments, and only OVA/LPS (presumably neutrophilic, non-type 2, etc.) was applied in more sophisticated studies (e.g. those presented in Figure 3 ) in this work? Why? They correspond to very different types of asthma?
  5. Besides, I would like to see for the experiments some data of which are demonstrated in Figure B and C (and whenever applicable) BALF levels of inflammatory cells, especially eosinophils and neutrophils to see that both models worked.
  6. In addition, in Figure 3F, it seems that in this presumably neutrophilic model, both neutrophils and eosinophils got increased in BALF as if it was a mixed model. Why?
  7. Was metabolomic performed in plasma? Please, make it clear whenever applicable.
  8. PBMCs were used instead of isolated pure cell populations, which is a limiting factor, especially in expression and/or epigenetic analyses. Please, discuss as a limitation of the study.
  9. By this occasion, considering the role of epigenetics in asthma-underlying pathobiology of airway epithelial cells (PMID: 31904412, 32973742), please, speculate on the potential role of epigenetic mechanisms is the findings reported by you’re here, even if not investigated.
  10. Besides, if your focus is airway epithelium, why PBMCs were studied?
  11. Clinical characteristics of the study subjects is rather good. Except for missing information on atopic sensitization. The data on specific serum IgE and/or skin prick testing should be added.
  12. Were the any differences in the data observed between atopic and non-atopic patients?
  13. You mentioned metabolic disturbances in obesity-associated asthma and that further studies should focus on the interaction between asthma and metabolic disorders. If However, some ways how the latter could affect the former have been identified, such as adipokines (PMID: 34948451) or extracellular vesicles (PMID: 37067460).
  14. What post-hoc test was used following ANOVA?
  15. Names of genes should be verified at https://www.ncbi.nlm.nih.gov/gene/ in a species-specific manner. They should be written in italics everywhere, including tables and figures.
  16. All abbreviations used in the figures must be explained in the respective legends.
  17. Please, try to improve the esthetics of the figures. For example, some titles of Y-axes are in italics while some other not, some lost formatting, etc.

Round 2

Reviewer 1 Report

Comments and Suggestions for Authors

Some additional improvements are necessary in the added text.

lines 317-332 -  the accuracy of the mean and SD should be the same. In biological measurement, three numbers are more than enough.

Author Response

comment1、lines 317-332 - the accuracy of the mean and SD should be the same. In biological measurement, three numbers are more than enough.

response、We thank the reviewer for this insightful comment. In response, we have revised the data formatting throughout the Results section to improve clarity and consistency. Specifically, all numerical data have been standardized to two decimal places across the manuscript. We sincerely appreciate this suggestion, which has contributed to enhancing the overall quality and readability of our data presentation.

Reviewer 3 Report

Comments and Suggestions for Authors

The authors addressed all comments 

Author Response

Comment:The authors addressed all comments 

Response: We sincerely thank the reviewer for their thorough review and are pleased to know that all concerns have been addressed satisfactorily.

Reviewer 5 Report

Comments and Suggestions for Authors

Thank you fir addressing my comments quite well.

This time, I have three minor points only:

  1. I am sorry, I was not accurate in my comment "Besides, I would like to see for the experiments some data of which are demonstrated in Figure B and C (and whenever applicable) BALF levels of inflammatory cells, especially eosinophils and neutrophils to see that both models worked.".
    Actually, I thought about Figures 2B and 2C. Please, provide to the corresponding experiment (shown in Figures 2B and 2C) graphs analogous to those shown for the other experiment if Figures 3E and 3F. Data shown in suppl table (file "table1-data for figure 3C.xlsx") are not required.
  2. You wrote "We thank the reviewer for this valuable comment. We agree that certain mechanisms linking metabolic disorders and asthma have already been identified, such as the involvement of adipokines [PMID: 34948451] and extracellular vesicles [PMID: 37067460]. In the revised manuscript, we have incorporated this information to better reflect current knowledge while also emphasizing the need for further studies to fully elucidate the complex interplay between asthma and metabolic dysregulation (line 71-78 in the manuscription of mark-up).
    However, I cannot see this what you describe in lines 71-78 of the marked manuscript? Another location (lines?)?
  3. What is reference "<bf02796504.pdf>." (reference 32)? 

Author Response

Comment1I am sorry, I was not accurate in my comment "Besides, I would like to see for the experiments some data of which are demonstrated in Figure B and C (and whenever applicable) BALF levels of inflammatory cells, especially eosinophils and neutrophils to see that both models worked.".

Actually, I thought about Figures 2B and 2C. Please, provide to the corresponding experiment (shown in Figures 2B and 2C) graphs analogous to those shown for the other experiment if Figures 3E and 3F. Data shown in suppl table (file "table1-data for figure 3C.xlsx") are not required.

Response1:We thank the reviewer for the clarification regarding Figures 2B and 2C. As part of our preliminary exploratory experiments, the mouse samples used in these figures were obtained from previously established models within the same experimental group. To address concerns about potential data reuse and to further validate the success of the model, we have now included additional data and explanation in the supplementary materials. These supplementary figures provide analogous BALF inflammatory cell profiles (eosinophils and neutrophils), similar to those shown in Figures 3E and 3F, to demonstrate that both models were successfully established.

Comment2You wrote "We thank the reviewer for this valuable comment. We agree that certain mechanisms linking metabolic disorders and asthma have already been identified, such as the involvement of adipokines [PMID: 34948451] and extracellular vesicles [PMID: 37067460]. In the revised manuscript, we have incorporated this information to better reflect current knowledge while also emphasizing the need for further studies to fully elucidate the complex interplay between asthma and metabolic dysregulation (line 71-78 in the manuscription of mark-up).

However, I cannot see this what you describe in lines 71-78 of the marked manuscript? Another location (lines?)?

Response2:We sincerely apologize for the oversight in referencing the incorrect line numbers. We updated the relevant background information regarding metabolic reprogramming and asthma in lines 41-48 of the marked-up manuscript. In the rationale for this study, we had reviewed the literature on lipid metabolism and its role in asthma. Therefore, in the revised Introduction section, we briefly introduced the current understanding of lipid metabolic dysregulation in asthma(reference 7-8,9-10) and emphasized the close relationship between metabolic reprogramming and asthma pathogenesis.

Comment3What is reference "<bf02796504.pdf>." (reference 32)?

Response3:We thank the reviewer for this important observation. We apologize for not citing the references in the correct format during the previous revision. This issue has been corrected in the revised manuscript.